# Principled approach to the selection of the embedding dimension of networks

Weiwei Gu[1], Aditya Tandon[2], Yong-Yeol Ahn[2,3,4] & Filippo Radicchi [2✉]

Network embedding is a general-purpose machine learning technique that encodes network structure in vector spaces with tunable dimension. Choosing an appropriate embedding dimension – small enough to be efficient and large enough to be effective – is challenging but necessary to generate embeddings applicable to a multitude of tasks. Existing strategies for the selection of the embedding dimension rely on performance maximization in downstream tasks. Here, we propose a principled method such that all structural information of a network is parsimoniously encoded. The method is validated on various embedding algorithms and a large corpus of real-world networks. The embedding dimension selected by our method in real-world networks suggest that efficient encoding in low-dimensional spaces is usually possible.

[1] UrbanNet Lab, College of Information Science and Technology, Beijing University of Chemical Technology, Beijing, P. R. China. [2] Center for Complex Networks and Systems Research, Luddy School of Informatics, Computing, and Engineering, Indiana University, Bloomington, IN, USA. [3] Network Science Institute, Indiana University, Bloomington (IUNI), IN, USA. [4] Connection Science, Massachusetts Institute of Technology, Cambridge, MA, USA. ✉email: filiradi@indiana.edu

Neural embedding methods are machine learning techniques that learn geometric representations of entities. For instance, word embedding methods leverage the relationships between words captured in large corpora to map each word to a vector[1–3]; graph embedding methods map each node to a vector by capturing structural information in the graph[4–7]. Embedding approaches have not only been pushing the performance envelope in many tasks, such as the prediction of word analogies and network links, but also provide a novel way to capture semantic and structural relationships geometrically[3,8–11].

In neural embedding of networks, the embedding space and dimension do not have a special meaning as in traditional embedding methods such as Laplacian Eigenmaps[12], or hyperbolic space embedding[13–17]. Instead, the dimension is considered as a hyperparameter of the model, which is either optimized through a model selection process or simply chosen based on the common practice (e.g., 100, 200, or 300 dimensions).

In word embedding, because we expect that the semantic space of human language would not drastically vary across corpora or languages, using common default parameters is reasonable, and the behavior of embedding models with the dimension parameter is rather well studied. For instance, it is common to use 100 to 300 dimensions, which are known to provide excellent performance in various tasks without a lot of over-parametrization risk[6,18–21]. By contrast, the space of graphs that we have is vast and we expect that there is no strong universal structure that may lead to similar optimal hyperparameters. Moreover, it is unclear how the structural properties of networks, such as community structure, would affect the right dimension of the model. For instance, imagine a road network, which is naturally embedded in a two-dimensional space. Because its geometrical nature, a suitable value for the embedding dimension should be close to two. Now, imagine another network that consists of two densely connected clusters of nodes, and only a few connections are present between the clusters. If there is little structural difference between the nodes within the same cluster, then even one dimension would suffice to represent the nodes effectively. Although the embedding dimension is one of the most important hyperparameter of the embedding models, particularly in graph embedding literature, it is difficult to find principled approaches to the proper selection of this hyperparameter value. Most existing methods use performance in downstream tasks, such as node classification and link prediction, to perform the model selection rather than establishing direct connections between the embedding dimension and the structural properties of the network. However, the performance for different tasks may be optimized with a different number of dimensions. Furthermore, it is natural to expect that the total information content of a network dataset is task independent.

Self-contained approaches to the selection of the embedding dimension of a network have been considered in previous papers. Classical embedding methods based on the spectral decomposition of the adjacency matrix or other graph operators, for example, take advantage of features of the spectrum of the operator to select the appropriate number of eigencomponents required to represent the graph with sufficient accuracy[12,22,23]. Other approaches are based on the underlying assumption that the network under observation is compatible with a priori given generative model defined in the geometric space. Then, the goodness of the fit of the observed network against the model provides an indication of the intrinsic dimension of the network itself. Irrespective of the specific model considered, the common message is that low-dimensional spaces are generally sufficient to obtain an accurate embedding of a network. In network embedding in Blau space for instance, the number of dimensions required to appropriately represent a network is expected to grow logarithmically with the network size[24–26]. Empirical analyses of large-scale social networks are in support of this claim[26]. Also, Hoff et al. considered generative network models where the probability of connection between pairs of nodes is a function of their distance in the embedding space[27]. They developed nearly-optimal inference algorithms to perform the embedding, and show that some small-size social networks can be quasi-optimally represented in three dimensions. We stress that expecting a low-dimensional embedding to be able to well represent a network does not mean that the network information is fully and exactly encoded by the embedding. This fact was emphasized by Seshadhri et al. who considered generic models where the probability of connection between pairs of nodes is a function of the dot product between the vector representations of the nodes in the embedding space[28]. The stochastic block model used in community detection[29–31] and the random dot product model[32,33] belong to this class of models. They mathematically proved that low-dimensional embeddings cannot generate graphs with both low average degree and large clustering coefficients, thus failing to capture significant structural aspects of real-world complex networks. The result seems, however, dependent on the specific model considered, as a minor relaxation of the model by Seshadhri et al. leads to exact low-dimensional factorizations of many real-world networks[34].

Here, we contribute to the literature on the subject by proposing a principled technique to choose a proper dimension value of a generic network embedding, and examine the relationships between structural characteristics and the chosen value of the embedding dimension. Our method does not aim at identifying the intrinsic dimension of a given network, rather the proper value of the space dimension of a given network according to a given embedding algorithm. Different embedding models applied to the same network may lead to different values of the selected embedding dimension. To this end, we follow a route similar to the recent study on word embedding dimension by Yin et al.[20]. Yin et al. proposed a metric, Pairwise Inner Product (PIP) loss function, that compares embeddings across different dimensions by measuring pairwise distances between entities within an embedding. By using this metric, they argued that a suitable value for the embedding dimension can be identified as the one corresponding to the optimal balance between bias and variance. Here, we extend the approach to the embedding of networks by proposing an alternative metric to quantify the amount of information shared by embeddings of different dimensions. We show that the content of structural information encoded in the embedding saturates in a power-law fashion as the dimension of the embedding space increases, and identify the proper value of the embedding dimension as the point of saturation of our metric. We evaluate our method by employing multiple embedding techniques, a host of real-world networks, and downstream prediction tasks.

## Results

**Embedding dimension and performance in detection tasks**. To compare embeddings of the same network obtained for different values of the embedding dimension $d$, we use the normalized embedding loss. We first set a reference dimension $d_r$, which we consider sufficiently large to capture the information stored in the original network. We then compute the embedding matrix $V^{(d)}$ for $d < d_r$ and use Eq. (6) to evaluate the normalized embedding loss function at dimension $d$ as $L(d) := L(V^{(d)}, V^{(d_r)})$. Specifically, for graphs with less than $N = 500$ nodes, we set the reference value $d_r = N$ and for graphs with more than $N = 500$ nodes, we set $d_r = 500$. The latter choice is made for convenience to avoid computational issues. Also, we stress that the choice of the

reference value $d_r$ is not so crucial as long as $d_r$ is large enough (Supplementary Fig. 1).

We stress that some embedding algorithms contain stochastic components (e.g., due to random sampling procedures), so that the embedding $V^{(d)}$ obtained for a given dimension $d$ may be not the same over different runs of the algorithm on the same network. We neglect this fact in the analysis performed in this section. We will illustrate how fluctuations affect our method in the selection of the embedding dimension below.

In the above recipe, we assume that the geometric representation obtained for $d = d_r$ is the most accurate representation that the embedding algorithm can obtain for the network. Such an assumption is obviously true for spectral algorithms of dimensionality reduction such as LE in $d_r = N$ dimensions, since the network topology can be exactly reconstructed from the full spectral decomposition of the normalized graph Laplacian. Any LE embedding in $d < N$ dimensions is necessarily a lossy representation of the network. Particularly for LE, the amount of topological information lost by suppressing some eigencomponents is a function of the eigenvalues of the normalized graph Laplacian. For other non-spectral embedding methods, similar assumptions should hold. The best geometric representation of a network, ideally a lossless representation, that an embedding method can achieve requires a space of dimension equal to the size of the network itself. The quality of a lower-dimensional embedding is quantified in terms of the information lost with respect to the most accurate embedding achievable by the embedding method on the network at hand.

An example of the application of our method is reported in Fig. 1a for the network of the American college football. Here, we use node2vec to obtain the embeddings. The reference value for this graph is $d_r = 115$, where $N = 115$ is the number of nodes.

As $d$ increases, $L(d)$ smoothly decreases toward an asymptotic value close to zero. This fact indicates that, as we increase the dimension $d$ of the embedding, the resulting geometric representations become similar to the one obtained for the reference value $d_r$. The saturation at high values of the embedding dimension is not due to degeneracy of the cosine similarity metric (Supplementary Fig. 2). As the embedding dimension grows the distribution of pairwise cosine similarities converges to a stable distribution with finite variance. Note that $L(d)$ is already very close to the asymptotic value for $d \simeq 10$, suggesting that the representation obtained by node2vec at $d = 10$ is already able to capture the structural features (pairwise distance relationships) of the network from the perspective of node2vec. Of course, this observation does not mean that 10 dimensions are sufficient to fully describe the observed network. It simply tells us that increasing the dimension $d$ of the embedding space is superfluous, in the sense that it does not significantly augment what node2vec is able to learn about the network.

This statement is further supported by our experiments on the performance in link prediction obtained for different values of the embedding dimension $d$, as illustrated in Fig. 1c. The accuracy in predicting missing links based on $d$-dimensional embeddings behaves almost exactly as $L(d)$. Increasing $d$ is useful up to a certain point. After that, there is no additional gain in prediction accuracy. The saturation of the link prediction accuracy arises earlier than the one observed for $L(d)$. This fact may be expected as there should be potentially more information in a geometric embedding than the one actually used in the link prediction task.

The analysis of the Cora citation network allows us to draw similar conclusions (Fig. 1b). Still, we see that $L(d)$ quickly decreases to an asymptotic value as $d$ increases. Furthermore, it seems that $d \simeq 45$ dimensions are more than enough to provide a

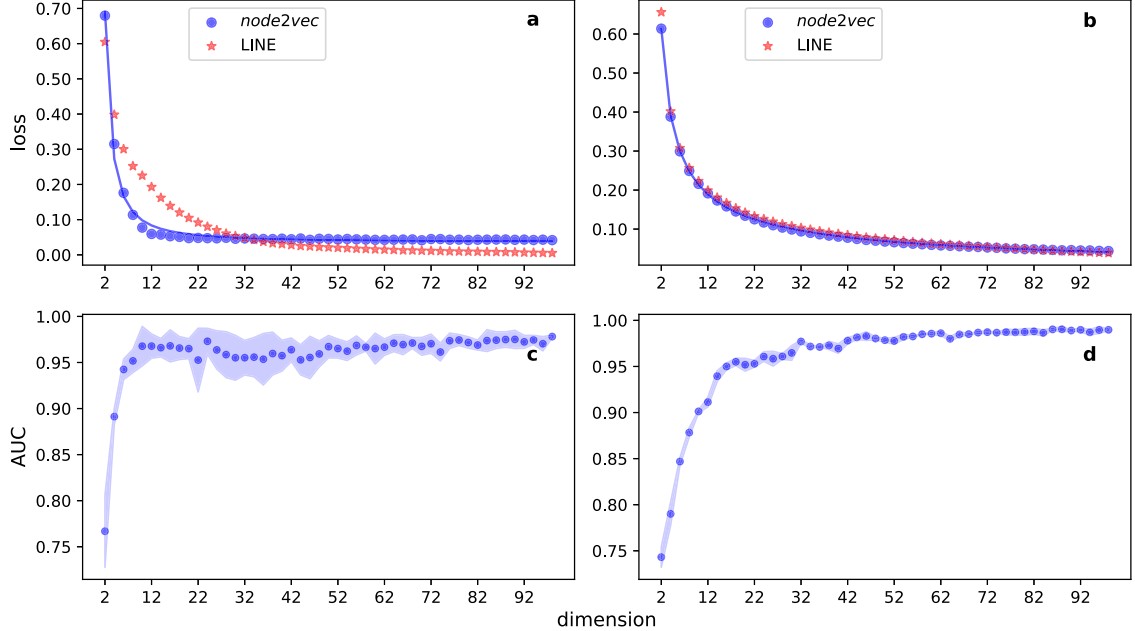

**Fig. 1 Geometric embedding of real-world networks. a** Normalized embedding loss as a function of the embedding dimension for the American college football network. Blue circles refer to numerical results obtained from node2vec; red stars refer to embeddings obtained with the LINE algorithm; the blue curve represents the best fit of Eq. (2) with the data points obtained with node2vec embeddings. We find $\hat{s} = 1.806$, $\hat{\alpha} = 1.468$ and $\hat{L}_\infty = 0.036$. The good quality of the fit is testified by the fact that the mean-squared error $R^2 = 4.252 \times 10^{-4}$. **c** AUCROC score of link prediction with the node2vec embedding as a function of the embedding dimension for the American college football network. Symbols represent average values of the AUCROC score over 10 random sub-sampling validation tests, while the shaded region identifies values within one standard deviation away from the mean. **b** Same as in (**a**), but for the Cora citation graph. The best fit of the data points with Eq. (2) is obtained for $\hat{s} = 1.030$, $\hat{\alpha} = 0.801$ and $\hat{L}_\infty = 0.048$ (mean-squared error $R^2 = 1.113 \times 10^{-4}$). **d** Same as in panel c, but for the Cora citation graph. A similar analysis has been performed for other real-world networks, see Supplementary Fig. 11.

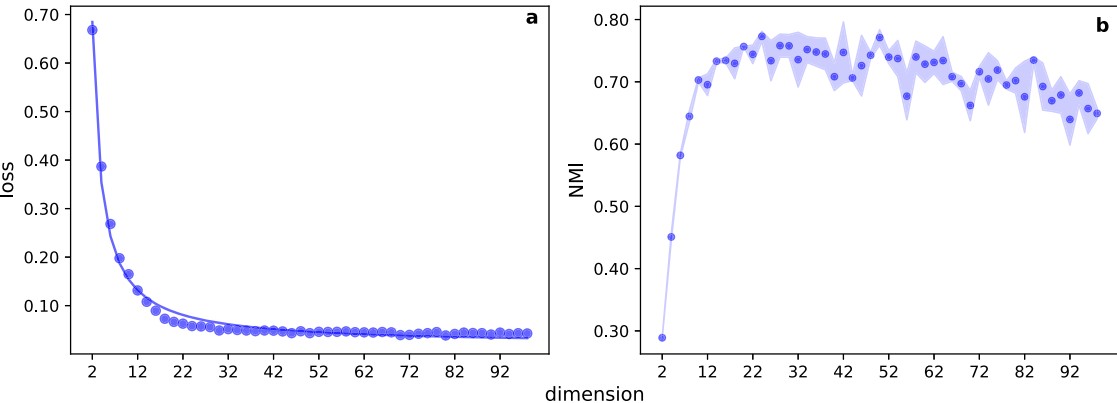

**Fig. 2 Geometric embedding of synthetic networks. a** Normalized loss as a function of the embedding dimension in SB networks. We generate graph instances composed of $N = 256$ nodes and $C = 16$ groups with $p_{in} = 0.2$ and $p_{out} = 0.02$. Embedding is performed using `node2vec`. The blue curve in panel a is the best fit curve of Eq. (2) ($\hat{s} = 1.323$, $\hat{\alpha} = 0.990$, $\hat{L}_\infty = 0.019$ and $R^2 = 5.93 \times 10^{-3}$) with the data points. **b** Test of performance in recovering the ground-truth community structure in SB graphs as a function of the dimension of the embedding. We measure the recovery accuracy with normalized mutual information (NMI). Symbols refer to average values of NMI computed over 10 instances of the SB model; the shaded region identifies the range of values corresponding to one standard deviation away from the mean.

sufficiently accurate geometric representation of the network that contains almost all the structural features that `node2vec` is able to extract from it. Also, the behavior of $L(d)$ predicts pretty well how the accuracy in link prediction grows as a function of the embedding dimension $d$ (Fig. 1d).

If we repeat the same analysis using the `LINE` algorithm[35], although the rate of decay of the function $L(d)$ is not identical for the two embedding algorithms, we find analogous behavior with $L(d)$ smoothly decaying toward an asymptotic value as $d$ increases (Fig. 1a, b). However, a faster decay doesn't necessarily imply that the corresponding algorithm is able to represent more efficiently the network than the other algorithm. The reference embeddings used in the definition of $L(d)$ are in fact algorithm dependent, and the quantitative comparison between the $L(d)$ functions of two different algorithms may be not informative.

Similar observations can be made when we perform the geometric embedding of synthetic networks. In Fig. 2, we display $L(d)$ for networks generated according to the SB model. We still observe a saturation of the function $L(d)$. Graph clustering reaches maximal performance at slightly smaller $d$ values than those necessary for the saturation of $L(d)$, and slowly decreases afterwards. The drop of performance for values of the embedding dimension that are too large has been already observed in other papers, see for example refs. [35,36]. Results based on `LE` embedding provide similar insights (Supplementary Figs. 3–6): $L(d)$ saturates quickly as $d$ increases; the performance in the graph clustering task is optimal before the saturation point of $L(d)$; further, the saturation point is compatible with the presence of a significant gap in the sorted Laplacian eigenvalues.

As we already mentioned it, we tested the robustness of our findings for different choices of the value of the reference dimension $d_r$ (Supplementary Fig. 1). We find that as long as $d_r$ is large enough, then the function $L(d)$ is basically unaffected by the specific choice made. Furthermore, we compared our loss function with the PIP loss function[20] (Supplementary Fig. 7). The functions behave in a qualitatively similar manner, displaying a clear decay toward an asymptotic value as the embedding dimension increases. The normalized loss function introduced here has the advantage of looking smoother than the PIP loss function, thus allowing for a mathematical description that requires less parameters.

Finally, we fed the k-means algorithm with `GraphSAGE` embeddings to detect the community structure of some real-world networks (Supplementary Fig. 8). We found that the

identified partition of a network does not significantly change if the network is embedded in a dimension larger than a certain threshold, and that such a threshold value is well predicted by the point of saturation of the normalized loss function associated with the `GraphSAGE` embeddings of the network. Similarly, node classification tasks based on `GraphRNA` embeddings reach stable performance if the dimension of the embedding is larger than a threshold (Supplementary Fig. 9). Such a threshold is well localized as the point of saturation of the normalized loss function of the `GraphRNA` network embeddings.

**Principled selection of a suitable value for the embedding dimension.** The observed behavior of the loss function $L(d)$ indicates that embedding algorithms may generate sufficiently accurate geometric descriptions with a relatively small value of $d$. Assuming that the plateau value $L_\infty$ of the loss function $L(d)$ corresponds to the best geometric description that the embedding algorithm can achieve, we indicate with

$$d_o(\epsilon) = \arg \min_d \left( L(d) - L_\infty < \epsilon \right), \qquad (1)$$

the minimal $d$ value such that the difference between $L(d)$ and the optimum $L_\infty$ is less than $\epsilon$. As already stated, a key implicit assumption here is that the empirically trained max-dimensional embedding is the best achievable by the algorithm. $d_o(\epsilon)$ is then the best choice for the embedding dimension that can be made to achieve a geometric representation that uses less training time and computational resources but is still sufficiently similar (i.e., within an $\epsilon$ margin) to the best achievable representation of the network. Our formulation of the problem does not explicitly account for the fact that the network at hand may be a specific realization of some underlying stochastic model which the embedding method may rely on. Rather, Eq. (1) defines a low-rank approximation problem, meaning that we aim at finding the low-dimensional (i.e., low-rank) approximation that best describes the high-dimensional (i.e., full-rank) embedding matrix. We differentiate from the standard setting considered in low-rank approximation problems in two main respects. First, we do not rely on a standard metric of distance (e.g., Frobenius distance) between the low- and the full-rank embedding matrices. Instead, we make use of a measure of collective congruence between the embeddings, i.e., Eq. (6). The normalized embedding loss function is equivalent to a metric of distance between matrices only for spectral embedding methods such as `LE`. However, the

equivalence is not immediate for arbitrary embedding methods. Second and more important, in a standard low-rank approximation, one generally aims at finding the best low-rank reduction of a full-rank matrix by constraining the maximum rank of the approximation. In our case instead, we seek the best rank reduction constrained by a maximum amount of tolerated discrepancy between the low- and the high-dimensional embeddings.

There could be several ways of finding the solution of Eq. (1). Here, given the smoothness of the empirically observed loss functions, we opted for a parametric approach. Specifically, we found that

$$L(d) = L_\infty + \frac{s}{d^\alpha}, \tag{2}$$

where $s$ and $\alpha$ are fitting parameters, represents well the data in various combinations of embedding methods and embedded networks. Equation (2) is meant to describe the mathematical behavior of the loss function only in the range $d \in [1, d_r]$. In particular in the definition of Eq. (2), $L_\infty$ appears as the value of the normalized loss function in the limit of infinitely large embedding dimensions. However, such a limit has no physical meaning and is used only to facilitate the mathematical description of the empirical loss function in the regime $d \ll d_r$. Best fits, obtained by minimizing the mean-squared error, of the function of Eq. (2) are shown in Figs. 1a, b and 2a. We performed a systematic analysis on a corpus of 135 real-world networks. We found that best estimates $\hat{L}_\infty$ of the asymptotic value $L_\infty$ for the normalized embedding loss function are generally close to zero (Supplementary Table 1). Best estimates $\hat{s}$ of the factor $s$, regulating the rate of the power-law decay toward $L_\infty$, seem very similar to each other, irrespective of the network that is actually embedded. Measured values of $\hat{s}$ indicate only a mild dependence on the underlying network (see Fig. 3). For uncorrelated clouds of points in $d$ dimension, the central limit theorem allows us to predict that $L(d) \sim 1/d^{1/2}$. Our best estimates $\hat{\alpha}$ of the decay exponent $\alpha$ are generally greater than those expected for uncorrelated clouds of data points, indicating that the embedding

algorithm correctly retains network structural information even in high-dimensional space. In systematic analyses performed on random networks constructed using either the ER and the BA models, we find that the size of the network is not an important factor in determining the plateau value $L_\infty$. The decay exponent $\alpha$ and the rate $s$ are positively correlated with density of the embedded network, but their values become constant in the limit of large network sizes (see Fig. 4).

Assuming the validity of Eq. (2), the solution of Eq. (1) for $d_o(\epsilon)$ can be written as

$$d_o(\epsilon) = \left(\frac{s}{\epsilon}\right)^{1/\alpha}. \tag{3}$$

The best estimate $\hat{d}_o(\epsilon)$ is calculated using the knowledge of the best estimates $\hat{s}$ and $\hat{\alpha}$ as $\hat{d}_o(\epsilon) = \left(\hat{s}/\epsilon\right)^{1/\hat{\alpha}}$. Values $\hat{d}_o(\epsilon)$ measured in real-world networks for $\epsilon = 0.05$ are reported in the Supplementary Table 1. In Fig. 3, we display the cumulative distribution of $\hat{d}_o(\epsilon = 0.05)$ over the entire corpus. For all networks in our dataset, we find that $\hat{d}_o(\epsilon = 0.05) \leq 218$. The size $N$ of the network is an upper bound for $d_o(\epsilon)$. However for sufficiently large networks, estimated values of $\hat{d}_o(\epsilon)$ do not display any clear dependence on $N$. Specifically, for roughly 40% of the real networks $\hat{d}_o(\epsilon = 0.05)/N < 0.01$, and for 80% of the real networks, $\hat{d}_o(\epsilon = 0.05)/N < 0.1$ (see Fig. 3).

The definition of the loss function provided in Eq. (6) uses cosine similarity to compare pairs of embeddings, tacitly assuming that the metric well represents distance of nodes in the embedding space. However, one may argue that this is not the case[28], and the outcome of the analysis may depend on the specific metric of distance adopted. For example, if the embedding is obtained by fitting an observed network against a generative network model, then the exact definition of connection probability in the model as a function of the distance between points in the space may play a fundamental role in properly embedding a network[27]. We therefore tested the robustness of our findings against different choices for the distance metric in

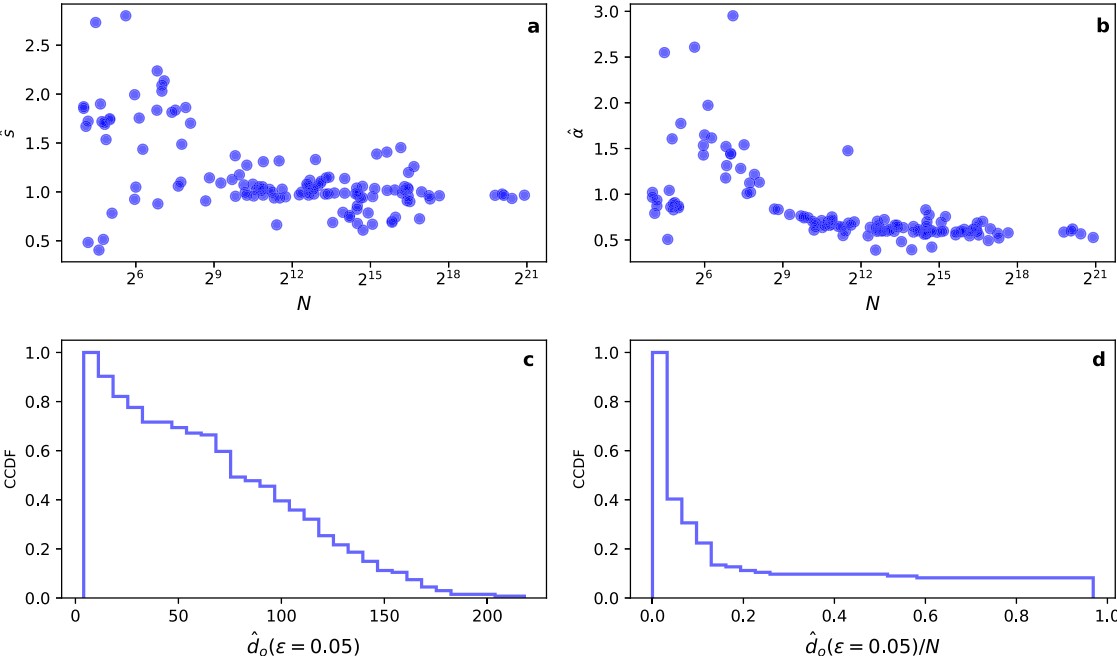

**Fig. 3 Principled selection of the embedding dimension of real-world networks. a** Distribution of $\hat{s}$ over the corpus of 135 real-world networks considered in this paper. **b** Same as in panel a, but for $\hat{\alpha}$. **c** Complementary cumulative distribution of $\hat{d}_o(\epsilon)$, with $\epsilon = 0.05$. **d** Complementary cumulative distribution of the best estimate $\hat{d}_o(\epsilon)$ rescaled by the network size $N$.

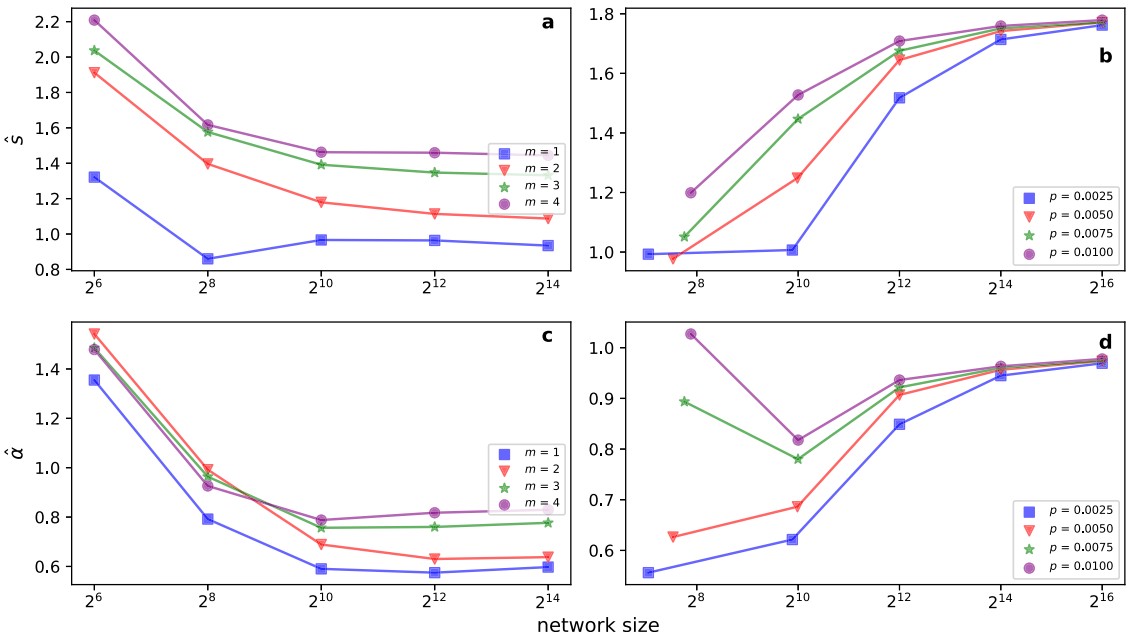

**Fig. 4 Dependence in the selection of the embedding dimension from network size and density. a** $\hat{s}$ as a function of the network size $N$ in the BA model. We control the network size and change network density of BA model by choosing different number of edges to attach from a new node to the existing nodes. **c** $\hat{\alpha}$ as a function of the network size for the same networks as in (**a**). **b**, **d** Same as in panels a and c, respectively, but for the ER model. Here, we control the density of the graphs by tuning the link probability $p$ between different nodes.

the embedding space (Supplementary Fig. 10). In our tests, we simply replace cosine similarity with another metric (e.g., Euclidean, correlation, and Chebyshev distances), and then look at how the normalized embedding loss changes as a function of the embedding dimension. We find that some differences are visible in small networks (e.g., the American college football); instead the curves obtained for large networks (e.g., Cora citation and Citeseer citation) are robust against the choice of the metric. We stress that the asymptotic value of the loss function is different from metric to metric, suggesting that some metric may be better at encoding pairwise similarity among nodes in the embedding space. This finding is perfectly in line with results by Hoff and collaborators[27]. However, the saturation of the content of information is similar irrespective of the actual distance metric adopted, supporting our main claims.

Similar conclusions can be made by looking at the values of $\hat{d}_o(\epsilon)$ obtained for node2vec and LE embeddings of synthetic networks with pre-imposed community structure (Supplementary Figs. 3–6). While there is a distinction between the actual value of the estimates $\hat{d}_o(\epsilon)$ for the two embedding methods, everything else is consistent. Specifically, the loss function $L(d)$ saturates as the embedding dimension increases. After the saturation, the performance in graph clustering doesn't improve anymore. For the LE embedding, the dimension value of the saturation roughly corresponds to the point where a neat gap between the eigenvalues of the graph normalized Laplacian is visible. The location of a neat gap among the eigenvalues of a graph operator is the standard approach for the identification of a suitable value for the embedding dimension in spectral dimensionality reduction methods. Our results indicate that the value of $\hat{d}_o(\epsilon)$ is not dependent on the specific metric used.

**Embedding dimension and variance of network embeddings.** Our definition of $d_o(\epsilon)$ of Eq. (1) corresponds to the minimal embedding dimension necessary to learn the structure of a network with a sufficient level of accuracy. However, we are not in the position to tell if the desired level of accuracy is reached

because the embedding algorithm is actually encoding the network structure in an optimal way, or instead the algorithm is over fitting the network. In this section, we perform a simple analysis in the attempt of providing some clarifications regarding this aspect of the problem.

We rely on the embedding coherence function of Eq. (7) to quantify the accuracy of a given algorithm to embed a network in $d$ dimensions. Specifically, we apply node2vec $K = 10$ times to the same network for every value of the embedding dimension $d$ to obtain a set of embeddings $V_d^{(1)}, \ldots, V_d^{(K)}$. Here, the diversity of the embeddings is due to the stochastic nature of the algorithm that relies on finite-size samples of random walks to embed the network. We then quantify the embedding coherence $S^2(d)$ of the algorithm at dimension $d$ using the embedding coherence function of Eq. (7) as $S^2(d) := S^2(V_d^{(1)}, \ldots, V_d^{(K)})$.

Figure 5 shows how $S^2(d)$ behaves as a function of $d$ for the same of the networks as we considered earlier. $S^2(d)$ is a non-monotonic function of $d$, showing a minimum value at a certain dimension $d$. We note that $S^2(d)$ decrease quickly toward its minimum, but it slowly increases afterwards. This finding concurs with previous observations that using higher-than-necessary dimensions does not critically affect the usefulness of the embeddings and may explain why the performance of embedding algorithms in tasks such as link prediction (Fig. 1) and graph clustering (Fig. 2) doesn't deteriorate as $d$ grows.

**Discussion**

In this paper, we proposed a principled solution to the problem of defining and identifying a suitable value of the embedding dimension of graphs. Our method is an extension to network data of the technique introduced by Yin et al.[20] in word embedding. The spirit of the approach is similar to the one adopted in spectral methods for dimensionality reduction, where a lossy representation of a network is obtained by suppressing an arbitrary number of eigencomponents of a graph operator (e.g., combinatorial and normalized Laplacians, adjacency matrix). The amount of information lost by neglecting some of the eigencomponents is a

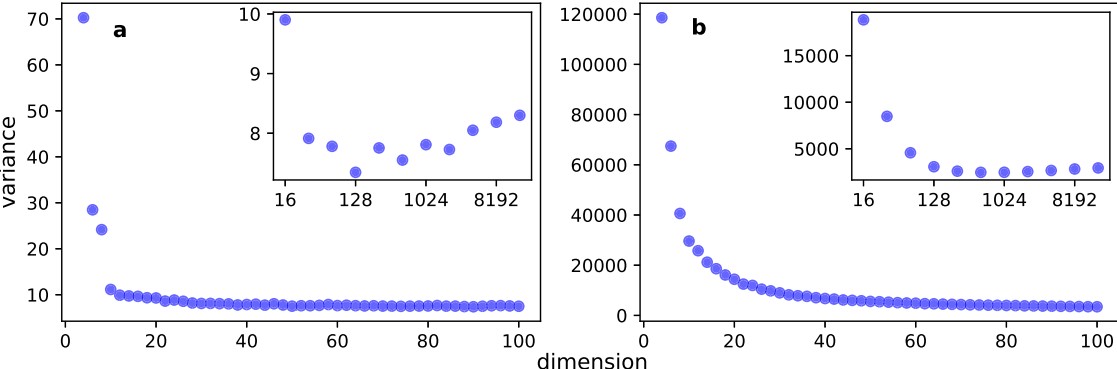

**Fig. 5 Embedding variance in real networks. a** Variance as a function of the embedding dimension $d$. Results are valid for `node2vec` embeddings of the American college football network. **b** Same as in panel a, but for the Cora citation network. A similar analysis has been performed for other real-world networks, see Supplementary Fig. 12.

function of the eigenvalues of the operator used in the reduction, and represents the main criterion to assess the effectiveness of the reduction itself. We generalize such an idea to an arbitrary graph embedding method. Accordingly, the effectiveness of an embedding in a low-dimensional space is measured as the ability to reproduce as closely as possible the best network representation that the embedding method can achieve. We assume that the best representation is obtained when the network is embedded in a space with dimension equal to the size of the network itself. We then measure the amount of information lost by reducing the dimension of the network representation. Finally, we identify a reasonable value of the embedding dimension, namely $d_o(\epsilon)$, as the smallest dimension value able to provide a sufficiently accurate (i.e., within an $\epsilon$ margin) representation of the network structure.

We validated the approach in two different ways. First, we applied it to embeddings based on the spectrum of the normalized graph Laplacian[12], and recovered values of $d_o(\epsilon)$ compatible with those obtained through well-established spectral criteria. Second, we validated the method by applying it to the cases where we could estimate the performance of embeddings in downstream tasks. We found that the embedding dimension value determined by our method roughly corresponds to the dimension value where the performance of standard tasks such as link prediction and graph clustering saturates to its maximum value. After validating it, we applied the method systematically to a large collection of real-world networks, finding that the estimated $\hat{d}_o(\epsilon)$ is much smaller than the number of nodes in the network.

In relation to the existing debate on the dimensionality of networks[28,34], we believe that our method may serve to provide an ex-post estimate for the network intrinsic dimension. If the loss function approaches zero for $\hat{d}_o = \hat{d}_o(\epsilon = 0)$, then it means that the exact representation of the network (i.e., the one valid for $d = N$) may be obtained in a $\hat{d}_o$-dimensional space. However, one cannot exclude the existence of a better method able to perfectly embed the same network in a space with $d < \hat{d}_o$ dimensions. Our systematic analysis of a corpus of 135 real-world networks bolsters the idea that the actual number of dimensions that are needed to describe the structure of a network is typically low[25–27].

## Methods

**Networks.** In this paper, we focus our attention on unweighted and undirected networks. The topology of a given network with $N$ nodes is described by its $N \times N$ adjacency matrix $A$. The entry $A_{ij} = A_{ji} = 1$ if a connection from node $i$ to node $j$ exists, whereas $A_{ij} = A_{ji} = 0$, otherwise.

*Empirical networks.* We consider a corpus of 135 network datasets. Sizes of these networks range from $N = 16$ to $N = 1,965,206$ nodes. We consider networks from different domains, including social, technological, information, biological, and

transportation networks (Supplementary Table 1). We ignore directions and weights of the edges, thus making every network undirected and unweighted. For illustrative purposes, we explicitly consider two real-world networks: the American college football network[37] and the Cora citation network[38]. The American college football network is a network composed of $N = 115$ nodes and $M = 613$ edges. Each node is a college football team. Two teams are connected by an edge if they played one against the other during the season of year 2000. The Cora citation network is composed of $N = 2,708$ nodes and $M = 5,429$ edges. Each node is a scientific paper and edges represent citations among papers.

*Network models.* In addition to the empirical networks, we consider three network generative models: the Erdős–Rényi (ER) model[39], the Barabási-Albert (BA) model[40], and the stochastic block (SB) model[29].

First, the ER model generates random networks with a Poisson degree distribution with average $\langle k \rangle = Np$. Here, $N$ is the total number of nodes and $p$ is the connection probability among pairs of nodes.

Second, the BA model is a growing network model that generates graphs obeying power-law degree distributions $P(k) \sim k^{-\gamma}$, with degree exponent $\gamma = 3$. To generate a single instance of the model, we specify the total number of nodes $N$ of the network, and the number of edges $m$ that each node introduces to the network during the growth process (the total number of edges in the network is $M = Nm$).

Finally, the SB model generates random networks with pre-imposed block structure, which solely determines the connection probabilities between nodes. Here, we implement the simplest version of the model, where $N$ nodes are divided into $C$ groups of identical size $N/C$. Pairs of nodes within the same group are connected with probability $p_{in}$; pairs of nodes belonging to different groups are connected with probability $p_{out}$. The total number of edges in the network is approximated by $M = N/Cp_{in} + (C - 1)N/Cp_{out}$.

**Network embedding algorithms.** The embedding in a $d$-dimensional space of a network with $N$ nodes is fully specified by the embedding matrix $V \in \mathbb{R}^{N \times d}$. The $i$-th row of the matrix $V$ is the vector $V_{i,\cdot}$ containing the coordinate values of node $i$ in the embedding. The entire procedure described in this paper can be applied to any embedding technique. Here, as a proof of concept and to ensure the robustness of the results, we consider five different embedding algorithms: `node2vec`[5], `LINE`[35], Laplacian Eigenmaps (`LE`)[12], `GraphSAGE`[19] and `GraphRNA`[18]. We center the embedding matrix $V$ so that $\sum_{i=1}^{N} V_{i,s} = 0$, for all $s = 1, ..., d$.

`node2vec`[5] is a popular network embedding algorithm that builds on the `word2vec` algorithm[3] by taking the following analogy: nodes in the network are considered as "words"; a sequence of nodes explored during a biased random walks is considered as a "sentence." In our analysis, we fix the number of walks per node to 20, the walk length to 10, the number of iterations to 10, and the parameters that bias the random walk toward a breadth-first or depth-first walk both equal to 1. The latter condition makes the embedding algorithm similar to `DeepWalk`[4].

`LINE`[35] is another popular embedding algorithm that aims at identifying embeddings that preserve first- and second-order neighborhood information. We use the default values for the algorithm parameters: the order of the proximity was set to 2, the number of negative samples to 5, and the initial learning rate to 0.025.

Laplacian Eigenmaps (`LE`)[12] is a classical embedding method based on the spectral decomposition of the graph Laplacian up to a desired eigencomponent. The importance of the various eigencomponents is inversely proportional to the magnitude of the corresponding eigenvalue (excluding the first trivial eigenvalue equal to zero), so that `LE` embedding up to a given dimension corresponds to

representing each node using the components of the ranked eigenvectors up to the given dimension value.

GraphSAGE[19] is an inductive framework for computing node embeddings in an unsupervised, supervised, or semisupervised way. Instead of training individual embeddings for each node, GraphSAGE learns a function that generates embeddings by sampling and aggregating features from the nodes' local neighborhoods. There are several ways for aggregating feature information from the nodes' neighbors. Here, we choose the mean aggregator (GraphSAGE-mean), which takes the element-wise mean of the representation vectors of a node's neighbors. GraphSAGE is based on the Graph Neural Network (GNN) framework. We interpret the number of hidden units in the last layer of the unsupervised GraphSAGE as the embedding dimension. In our experiments, we vary the number of units of the last layer while keeping other parameters unchanged.

GraphRNA[18] learns node representations for networks with attributes by feeding bidirectional Recurrent Neural Networks (RNNs) with random walk sequences. In our experiments, we apply the node classification task to evaluate its embedding performance. The number of neural units of the last layer of the bidirectional RNNs is set equal to the number of classes of the nodes in the network. All parameters of the embedding algorithm are kept unchanged, but we vary the number of hidden units in the second-last layer of the bidirectional RNNs. We interpret the number of units in the second-last layer as the embedding dimension.

**Task-based evaluation of network embedding methods**. We consider three quantitative evaluation tasks: link prediction, graph clustering and node classification. All tasks are considered as standard tests for the evaluation of the performance of graph embedding methods[7,18,41].

*Link prediction*. For the link prediction task, we use the repeated random subsampling validation by applying the following procedure 10 times. We report results corresponding to the AUCROC score (the area under a receiver operating characteristic curve) over these repetitions. We first hide 10% of randomly chosen edges of the original graph. The hidden edges in the original graph are regarded as the "positive" sample set. We sample an equal amount of disconnected vertex pairs as the "negative" sample set. The union of the "positive" and "negative" sample sets form our test set. The training set consists of the remaining 90% of connected node pairs and an equal number of randomly chosen disconnected node pairs. We use the training set to learn the embedding and determine the embedding of the nodes. Furthermore, we use the training set to learn the probability of connection between pairs of nodes given their embedding. Specifically, for every pair of nodes $i$ and $j$ in the training set, we evaluate the Hadamard product $e_{ij} = V_{i,\cdot} \odot V_{j,\cdot}$. Note that $e_{ij}$ is a $d$-dimensional vector. We assume that the probability of connection between nodes $i$ and $j$ is given by

$$p_{ij}(e_{ij}; \theta) = \frac{1}{1 + \exp(-e_{ij}^T \theta)} \qquad (4)$$

where $\theta$ is a $d$-dimensional parameter vector, and $e_{ij}^T \theta$ is the dot product between the vectors $e_{ij}$ and $\theta$[42]. The best estimate of the entries of vector $\theta$ are obtained from the training set via logistic regression. In absence of training when all components of the vector $\theta$ are identical and positive, the probability of Eq. (4) is proportional to the dot product between the vectors $V_{i,\cdot}$ and $V_{j,\cdot}$. We use Eq. (4) to rank edges of the test set and evaluate the AUCROC score of the link prediction task.

*Graph clustering*. We take advantage of the SB model to generate graphs with pre-imposed cluster structure. We consider various combinations of the model parameters, $N$, $C$, $p_{in}$ and $p_{out}$.

We then perform the embedding of the graph using information from its adjacency matrix only. To retrieve clusters of nodes emerging from the embedding, we use the k-means clustering algorithm[43]. In the application of the clustering algorithm, we provide additional information by specifying that we are looking for exactly $C$ clusters. The retrieved clusters of nodes are compared against the true clusters imposed in the construction of the SB model. Specifically, we rely on the normalized mutual information (NMI) to measure the overlap between the ground-truth clusters and the detected clusters[44]. Values of NMI close to one indicate an almost perfect match between the two sets; NMI equals zero if detected and ground-truth clusters have no relation.

Also, we apply graph clustering to real-world networks. We still identify clusters via the k-means algorithm applied to graph embeddings. The retrieved clusters of nodes are compared against the clusters identified by the popular community detection method Infomap[45]. In the application of the k-means algorithm, we specify that we are looking for a number of clusters equal to the number of communities found by Infomap. NMI is used is to quantify the level of similarity between the clusters identified by the two methods.

*Node classification*. In the node classification task, given the embedding matrix $V$ and the labels of all nodes, we first randomly select 90% of the node representations

from $V$ to form the training set. The remaining 10% of the nodes constitutes the test set. Within the training set, we randomly select 10% of the node representations to form the validation set. To conduct the node classification task, we leverage the training set and the corresponding labels to train a multilayer perceptron classifier and use the validation set to fine tune some hyperparameters. The micro average metric is used to quantify the accuracy of the node classification over the test set. The results of the node classification task reported in our paper are given by the arithmetic average over 10 independent runs of the above procedure.

**Evaluation metrics of network embeddings**. Here, we define two metrics for assessing the quality of network embeddings.

*Normalized embedding loss function*. We define a loss function similar to the Pairwise Inner Product (PIP) loss function used for word embeddings[20]. The metric can be used to compare any pair of embeddings, regardless of the way they are obtained, as long as they refer to the same network with the same set of nodes. The function takes two inputs $V^{(a)}$ and $V^{(b)}$, respectively representing the matrices of the embeddings $a$ and $b$ that we want to compare. We preprocess each of these matrices by calculating the cosine similarity between all node pairs. For the embedding matrix $V^{(a)}$, we compute

$$C(V^{(a)}) = V^{(a)} \left[V^{(a)}\right]^T. \qquad (5)$$

A similar expression is used to compute $C(V^{(b)})$ for the embedding matrix $V^{(b)}$. As we already stated, the embedding matrices $V^{(a)}$ and $V^{(b)}$ are appropriately centered to account for fact that cosine similarity is not a translation-invariant metric. $C(V^{(a)})$ is a $N \times N$ matrix that captures the pairwise similarity between nodes; $[C(V^{(a)})]_{i,j}$ corresponds to the cosine similarity between nodes $i$ and $j$ in the embedding $V^{(a)}$.

The normalized embedding loss function $L(V^{(a)}, V^{(b)})$ is defined as the average, over all possible node pairs, of the absolute difference values between the cosine similarity of the two embeddings, i.e.,

$$L(V^{(a)}, V^{(b)}) = \frac{2}{N(N-1)} \sum_{i<j} \left| [C(V^{(a)})]_{i,j} - [C(V^{(b)})]_{i,j} \right|. \qquad (6)$$

$L(V^{(a)}, V^{(b)}) = 0$ if the two embeddings $a$ and $b$ are equivalent. We expect instead $L(V^{(a)}, V^{(b)}) \simeq 2$ if the two embeddings represent two radically different representations of the network.

Cosine similarity is chosen for its simplicity. We verify that the use of other similarity/distance metrics in the normalized embedding loss function of Eq. (6) in place of cosine similarity provides outcomes qualitatively similar to those reported in the paper (Supplementary Fig. 10).

*Embedding variance*. Another metric that we use is "embedding variance," which estimates the level of coherence among a set of multiple embeddings of the same network. The metric takes a set $\{V^{(1)}, V^{(2)}, \ldots, V^{(K)}\}$ of $K$ embedding matrices of the same network as its input and calculates the average value of the variance of the node pair similarities across the embeddings. We obtain the cosine similarity matrix $C(V^{(k)})$ for each of the $k = 1, \ldots, K$ embeddings using Eq. (5). We compute the embedding variance as

$$S^2(V^{(1)}, V^{(2)}, \ldots, V^{(K)}) = \frac{1}{K} \sum_{k=1}^{K} \sum_{i<j} \left[ C(V^{(k)})_{i,j} - \langle C_{i,j} \rangle \right]^2, \qquad (7)$$

where

$$\langle C_{i,j} \rangle = \frac{1}{K} \sum_{k=1}^{K} C(V^{(k)})_{i,j}$$

is the average value of the cosine similarity of the nodes $i$ and $j$ over the entire set of embeddings. Equation (7) equals the variance of the cosine similarity over all pairs of nodes in all embeddings. $S^2(V^{(1)}, V^{(2)}, \ldots, V^{(K)}) = 0$, if the embeddings are such that the cosine similarity of all node pairs is always the same across the entire set of embeddings. High values of $S^2(V^{(1)}, V^{(2)}, \ldots, V^{(K)})$ indicate low coherence among the embeddings in the set. Note that, contrary to Eqs. (6) and (7) applies only to embeddings of the same dimension.

## Data availability
All network datasets considered in this paper have been obtained from http://konect.cc/networks/. We include complete information on how to obtain individual datasets in Supplementary Table 1.

## Code availability
The Python script used for this project is available online at https://zenodo.org/record/4757121#.YKKPLC-cZp8[46].

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

## Acknowledgements

W.W. acknowledges financial support from the National Science Foundation of China (61903020), the Swarma Club and the program of China Scholarships Council (No.201806040107). Y.Y.A. acknowledges support from the Air Force Office of Scientific Research (FA9550-19-1-0391). F.R. acknowledges support from the National Science Foundation (CMMI-1552487) and the US Army Research Office (W911NF-16-1- 0104).

## Author contributions

W.W., Y.Y.A., and F.R. wrote the paper. W.W., A.T., Y.Y.A., and F.R. designed the experiments. W.W. and A.T. performed the experiments.

## Competing interests

The authors declare no competing interests.
