## [Peer Review File · Nature Communications]

Reviewers' Comments:

Reviewer #2:

Remarks to the Author:

In the paper, the authors investigated dimension selection for network embedding algorithms. From empirical experiments conducted on several network embedding algorithms (node2vec, LINE and Laplacian Eigenmaps) on both real-word and generated graphs, the authors found that their loss criterion, the normalized embedding loss (which is an extension of the Pairwise Inner Product loss) can be used to identify optimal dimensions (within an ϵ -accuracy of the "optimal" embedding, which is assumed to be an infinite-dimension embedding). They also verified empirically, that other extrinsic metrics (e.g. link prediction and graph clustering) generally agrees with results using the normalized embedding loss. Despite having several flaws in the problem formulation, as a paper focusing on empirical results, the observations and discoveries of results on graph embedding dimension is rather interesting.

It has been noted that dimensionality is a crucial hyper-parameter in representation learning and deep learning. As a result, I appreciate the effort presented by the author on the investigation of this hyper-parameter for network embeddings. The findings are rather assuring, which seem to suggest that there is a plateau effect -- once the dimension is larger than a threshold (which is usually reasonable), the performance does not get a lot better onward. While I do agree that this is generally observed (but probably not yet systematically studied) by researchers, the notion of "optimality" in the paper is a bit concerning. When one assumes that the optimal model is the one with infinite capacity (in this case, embedding of very large dimensionality), there is a concern that the variance part (which causes overfitting) is ignored in the analysis.

Under this framework, the "optimal dimension" is only from the view of computation and model size, but not so much performance-wise. While in other related research, the "optimality" is actually in terms of performance -- larger or small dimensions will both lead to degraded models. For network embedding, this effect has also been discovered (e.g. [Ma, Yao, et al. "Multi-dimensional network embedding with hierarchical structure." Proceedings of the eleventh ACM international conference on web search and data mining. 2018.]). This phenomenon was also revealed in the original papers of LINE (where additional dimensions lead to worse scores). Plenty of evidences in graph embedding research have shown that this over-fitting effect exists, and it's a bit worrisome that the authors ignored this point completely -- in Figure 1 (c) and (d) the accuracy began to drop (although not dramatically) when dimension becomes too large, and Figure 2 (b) shows the same trend as well. They seem to suggest that infinite-dimensional embeddings are not in fact the optimal ones.

On a positive note, the experiments on real-world tasks (clustering and link prediction) show that embedding performance is not strongly affected by over-parametrization, and that the embedding sizes are more or less correct as long as it is not too small. Moreover, the dimensions achieving optimal performance are usually in accordance with real-world usages (a few 10's). I encourage the authors to expand this section by adding more experimentation results on more datasets and tasks. Not only are they supportive to the authors' arguments, they (as a systematic review of the effect of dimension on network embedding tasks) will also be useful for other researchers and practitioners.

The paper is well structured and is easy to follow. As a nitpicking, additional rounds of proofreading might be helpful as some minor grammatical issues do exist (e.g. "Because its geometrical nature" and "...take advantage of features of the spectrum"). Overall the paper has some merits (which the authors could have done better and more thorough, e.g. the experimentation on dimensionality for real-world embedding tasks), but it is flawed. In particular, the authors considered only the bias, but not the variance. It is possible to address the issues and I think a major revision is needed.

Reviewer #3:

Remarks to the Author:

The paper contributes to the important question - what is the optimal dimension for network embedding? In machine learning the answer to this question typically depends on an application at hand - the dimension is set sufficiently high for a particular application to work sufficiently well, but not much higher than that. The paper looks for a more principled approach. It adjusts a recently introduced loss function (ref.18) that measures similarity between two distance matrices. The loss function (eq.3) is a variation of the basic idea - a normalized difference between pairwise distances between embedded nodes. Using this loss function, distance matrices of embeddings of different dimensions are compared. The differences between distance matrices are negligible at very high dimensions, so that the optimal dimension is defined as the smallest dimension whose distance matrix is away by epsilon from a higher-dimensional distance matrix according to the loss function.

The optimal dimension of a network in this definition depends on the definitions of embedding distance, loss function and epsilon. It is shown in SI that the results are qualitatively (but not quantitatively) similar for different definitions of distances. The optimal dimension of many networks is found to be low (for epsilon=0.05 and dot-product distances), corroborating previous results of this type in machine learning. The main difference between how embedding dimension is determined in this paper and in a typical machine learning paper on the same subject is the shift of focus from application performance metrics to embedding similarity metrics.

This shift of focus is the source of main concern. The embedding algorithms featured in the paper are not based on any network models, so that the embedding space in these algorithms is truly virtual - it does not have any physical or network-modeling meaning by itself. It is an artificial construct useful only in applications. Therefore the search for a "true" embedding dimension, presented as the main motivation in the paper, appears misguided. However, it is shown that the proposed definition of optimal embedding dimension yields qualitatively reasonable results for link prediction and community inference applications.

Before further comments could be made concerning the paper, it needs to be revised to address the following questions.

1. It is understood in machine learning that the "saturation" at high dimensions is often due to degeneracy, not overparametrization. The difference between higher-dimensional distance matrices may be small because these matrices become degenerate, all the distances concentrating around the same value, the distance distribution becoming a delta function centered at this value. Concretely, the dot-product distances between embedded nodes may be all nearly zero at high dimension because all vectors are nearly orthogonal, Euclidean distances all becoming the same fraction of the diameter of the embedding space, and so on. The paper needs to show that the distance distributions at high dimensions are not trivially degenerate, not converging to delta functions.

2. Eq.6 used in the definition of the optimal embedding dimension does not look correct unless L_{∞} is exactly zero. This is because at $d=d_r$, $L(d)$ must be zero, but it is not zero according to eq.6. Also, what is $L(d)$ for $d \gg d_r \gg 1$? In any case, $L(d)$ converging to a not-exactly-zero constant at $d \rightarrow \infty$ is an unexplained mystery.

3. The link prediction part is very unusual in many respects, and needs to be either expanded with extensive justifications or, better, made conventional. The most objectionable part is to call link ij present if $p_{ij} > 0.5$. If this is how a network is constructed, then it is surely very different from the original network! If the path leading to p_{ij} is followed (which has its own many issues), then the standard practice is to rank all ij according to p_{ij} and report the standard metrics - AUPRC, AUC-ROC and similar.

Some other suggestions of less importance.

4. The discussion of normalization of $V^{a,b}$ is missing around eqs.2,3. If they are not normalized, the distances C_{ij} can exceed 1. If they are properly normalized, L cannot exceed 1 ("2" after eq.3 is then a typo).

5. It is not mentioned that eq.4, contrary to eq.3, applies only to embeddings of the same dimension.

6. A more representative set of citations in machine learning, other than 18,19, could be composed to support the claim that the embedding dimension of many networks is known to be low.

Reviewer #2

In the paper, the authors investigated dimension selection for network embedding algorithms. From empirical experiments conducted on several network embedding algorithms (node2vec, LINE and Laplacian Eigenmaps) on both real-word and generated graphs, the authors found that their loss criterion, the normalized embedding loss (which is an extension of the Pairwise Inner Product loss) can be used to identify optimal dimensions (within an ϵ -accuracy of the “optimal” embedding, which is assumed to be an infinite-dimension embedding). They also verified empirically, that other extrinsic metrics (e.g. link prediction and graph clustering) generally agrees with results using the normalized embedding loss. Despite having several flaws in the problem formulation, as a paper focusing on empirical results, the observations and discoveries of results on graph embedding dimension is rather interesting.

We appreciate the positive opinion about our manuscript, and thank the reviewer for the constructive and insightful comments that we addressed as follows.

It has been noted that dimensionality is a crucial hyper-parameter in representation learning and deep learning. As a result, I appreciate the effort presented by the author on the investigation of this hyper-parameter for network embeddings. The findings are rather assuring, which seem to suggest that there is a plateau effect – once the dimension is larger than a threshold value (which is usually reasonable), the performance does not get a lot better onward. While I do agree that this is generally observed (but probably not yet systematically studied) by researchers, the notion of “optimality” in the paper is a bit concerning. When one assumes that the optimal model is the one with infinite capacity (in this case, embedding of very large dimensionality), there is a concern that the variance part (which causes overfitting) is ignored in the analysis.

We thank the reviewer for making this comment. We revised several parts of the manuscript to better clarify the assumptions of our approach for the definition of optimal embedding dimension. Also, we included results from two additional graph embedding algorithms. Below, we briefly reply to the criticism by the reviewer.

As correctly stated by the reviewer, we tacitly assume that perfect embedding is possible for $d = N$. An example of such a perfect embedding (i.e., lossless representation) is trivially given by the adjacency matrix of the graph, with each row of the matrix containing the coordinates of the corresponding node in the space. Please note that we are not assuming that an (alternative) perfect embedding is not possible for $d < N$. However, we do not a priori know if such a perfect representation is possible for an arbitrary network.

A lossless representation is naturally an overfit as it serves to represent only the actual network at hand. A potential concern is present if the embedding is used to predict network properties that are not directly available to the observer. For example, it is natural to expect that a lossless

representation may not optimally serve in the prediction of missing links. However, no concern is present if the interest is in representing as accurately as possible the network in a lower-dimensional space. In this context, no prediction task is explicitly considered, and all network information is available to the observer.

To be more concrete, consider the analogy between our method and traditional spectral methods for dimensionality reduction. There, dimensionality reduction of a network is performed by suppressing an arbitrary number of eigencomponents of a graph operator (e.g., combinatorial or normalized Laplacians, adjacency matrix). The amount of information lost by neglecting eigencomponents is a function of the eigenvalues of the operator used in the reduction. We remark that the loss of information is relative to the specific operator considered in the spectral decomposition of the graph. Performance in downstream tasks depends on the operator considered too.

In our paper, we indeed included the method Laplacian Eigenmaps (LE) to create an explicit connection between our novel approach and the standard approach used in spectral methods for dimensionality reduction. Our numerical results demonstrate that our method identifies the network optimal dimension of the LE embedding in proximity of the spectral gap of the normalized graph Laplacian.

Further in our revised manuscript, we included results for the graph embedding algorithm GraphSAGE [1]. As shown in Fig. S10, once the embedding dimension is larger than some finite value, then the community structure detected through the embedding does not change. The stability in the outcome of the community detection task is achieved for a value of the dimension of the embedding space that is similar to one visible in the saturation of the normalized loss function. Saturation of performance in community detection with the dimension of the embedding space has been reported already for some embedding methods, see for example Refs. [3, 6, 4, 2, 5]. Saturation behavior appears also from the application of the embedding algorithm GraphRNA [3] in node classification tasks, see Fig. S11 of the revised version of the SI.

Finally, we had already reported on how the embedding variance changes as a function of the dimension of the embedding space, see Fig. 5 in the main text and Fig. S9 in the SI. We performed the analysis only for a small selection of networks embedded with node2vec. However, the results are pretty clear. Embedding variance displays a minimum for intermediate values of the embedding dimension. For larger values of the dimension, the embedding variance grows very slowly. The finding is in accordance with our previous observations that using higher-than-necessary dimensions does not dramatically affect the ability of the algorithm to embed the network.

Under this framework, the “optimal dimension” is only from the view of computation and model size, but not so much performance-wise. While in other related research, the “optimality” is actually in terms of performance – larger or small dimensions will both lead to degraded models. For network embedding, this effect has also been discovered (e.g. [Ma, Yao, et al. “Multi-dimensional network embedding with hierarchical structure.” Proceedings of the eleventh ACM international conference on web search and data mining. 2018.]). This phenomenon was also revealed in the original papers of LINE (where additional dimensions lead to worse scores). Plenty of evidences in graph embedding research have shown that this over-fitting effect exists, and it’s a bit worrisome that the authors ignored this point completely – in Figure 1 (c) and (d) the accuracy began to drop (although not dramatically) when dimension becomes too large, and Figure 2 (b) shows the same trend as well. They seem to suggest that infinite-dimensional embeddings are not in fact the optimal ones.

We agree with the reviewer that performance in downstream tasks may slightly deteriorate in the limit of large dimension values. This fact is apparent from many of our results, including link prediction and clustering. However, the drop in performance is not as dramatic. In the revised version of the manuscript, we included a few sentences to better emphasize this fact. We further included a reference to the paper by Ma *et al.*, as indicated by the referee.

At the same time, we remark that our definition of optimality does not concern performance in downstream tasks rather geometric congruence between the low- and high-dimensional embeddings of the network obtained by the specific embedding method at hand. This is the main conceptual novelty of our work. We do use downstream tasks to validate our definition of optimal embedding dimension and provide an explicit connection to the “standard” notion of optimality adopted in the literature. Specifically, we show that good performance in downstream tasks is obtained only if the embedding space has a dimension sufficiently large to obtain an embedding congruent with the one valid in high-dimensional space.

On a positive note, the experiments on real-world tasks (clustering and link prediction) show that embedding performance is not strongly affected by over-parametrization, and that the embedding sizes are more or less correct as long as it is not too small. Moreover, the dimensions achieving optimal performance are usually in accordance with real-world usages (a few 10’s). I encourage the authors to expand this section by adding more experimentation results on more datasets and tasks. Not only are they supportive to the authors’ arguments, they (as a systematic review of the effect of dimension on network embedding tasks) will also be useful for other researchers and practitioners.

We thank the reviewer for the very positive comment. We followed the advice and expanded the analysis in two main respects. First, we included some additional large-scale real-world networks in our analysis. Second, we applied our approach to graph embedding methods that rely on neural-network-based representations. Specifically, we consider GraphSAGE [1] and GraphRNA [3]. Results are reported in Fig. S10 for GraphSAGE and in Fig. S11 for GraphRNA. Both methods

display a saturation in information loss as the dimension of the embedding increases. A similar saturation is observed when the embedding methods are used in downstream tasks. We specifically consider, community detection for GraphSAGE and node classification for GraphRNA.

The paper is well structured and is easy to follow. As a nitpicking, additional rounds of proofreading might be helpful as some minor grammatical issues do exist (e.g. “Because its geometrical nature” and “...take advantage of features of the spectrum”). Overall the paper has some merits (which the authors could have done better and more thorough, e.g. the experimentation on dimensionality for real-world embedding tasks), but it is flawed. In particular, the authors considered only the bias, but not the variance. It is possible to address the issues and I think a major revision is needed.

We thank the reviewer for the time spent on our work. We appreciate the very constructive spirit of the criticisms. We believe that revised version of the manuscript is a significant improvement over the original version we submitted, and we hope that the referee will support its publication in the journal.

Reviewer #3

The paper contributes to the important question - what is the optimal dimension for network embedding? In machine learning the answer to this question typically depends on an application at hand - the dimension is set sufficiently high for a particular application to work sufficiently well, but not much higher than that. The paper looks for a more principled approach. It adjusts a recently introduced loss function (ref.18) that measures similarity between two distance matrices. The loss function (eq.3) is a variation of the basic idea - a normalized difference between pairwise distances between embedded nodes. Using this loss function, distance matrices of embeddings of different dimensions are compared. The differences between distance matrices are negligible at very high dimensions, so that the optimal dimension is defined as the smallest dimension whose distance matrix is away by epsilon from a higher-dimensional distance matrix according to the loss function.

We appreciate the positive opinion about our manuscript. We took advantage of the pertinent criticisms to improve our manuscript. We report below a point-to-point reply to all comments made the reviewer.

The optimal dimension of a network in this definition depends on the definitions of embedding distance, loss function and epsilon. It is shown in SI that the results are qualitatively (but not quantitatively) similar for different definitions of distances. The optimal dimension of many networks is found to be low (for epsilon=0.05 and dot-product distances), corroborating previous results of this type in machine learning. The main difference between how embedding dimension is determined in this paper and in a typical machine learning paper on the same subject is the shift of focus from application performance metrics to embedding similarity metrics.

Thanks, this is a good summary of the main results of the paper!

We remark that our definition of optimal dimension depends on the embedding method at hand too. The optimal dimension value is determined by solving the optimal trade-off between efficiency and effectiveness of the embedding. We indeed assume that efficiency is inversely proportional to the dimension of the embedding space; effectiveness is given by the ability of the embedding method to obtain, in a low-dimensional space, a representation of the network that is congruent with the one obtained in $d = N$ dimensions, with N network size. The parameter ϵ is used to control the minimum level of acceptable effectiveness.

This shift of focus is the source of main concern. The embedding algorithms featured in the paper are not based on any network models, so that the embedding space in these algorithms is truly virtual - it does not have any physical or network-modeling meaning by itself. It is an artificial construct useful only in applications. Therefore the search for a “true” embedding dimension, presented as the main motivation in the paper, appears misguided. However, it is shown that the proposed definition of optimal embedding dimension yields qualitatively reasonable results for link prediction and community inference applications.

We agree with the reviewer, and we appreciate the constructive spirit of the criticism. We attempted to improve the presentation of the main motivation of the paper by performing significant revisions to the main text.

We stress that our work is not aiming at identifying the intrinsic dimension of a given network, rather the optimal dimension of a given network according to a given embedding method. Different methods applied to the same network may lead to different values of the optimal embedding dimension. The quality of the embedding of a network is method/model dependent, and we do not have a way to measure directly the intrinsic dimension of a network. We can only provide an ex-post estimate of an upper bound of its intrinsic dimension. In fact, if the loss function approaches zero for $d_o < N$, then it means that the exact representation of the network (i.e., the one valid for $d = N$) can be obtained equally well in a d_o -dimensional space. However, we cannot exclude the existence of a better method able to perfectly embed the same network in a space with $d < d_o$ dimensions.

As the reviewer also remarks, our multiple tests indicate that the selected values of the embedding dimension allow for reasonable results in downstream tasks. So, in a way, they appear suitable for the generation of pretty good models of networks.

Before further comments could be made concerning the paper, it needs to be revised to address the following questions.

1. It is understood in machine learning that the “saturation” at high dimensions is often due to degeneracy, not overparametrization. The difference between higher-dimensional distance matrices may be small because these matrices become degenerate, all the distances concentrating around the same value, the distance distribution becoming a delta function centered at this value. Concretely, the dot-product distances between embedded nodes may be all nearly zero at high dimension because all vectors are nearly orthogonal, Euclidean distances all becoming the same fraction of the diameter of the embedding space, and so on. The paper needs to show that the distance distributions at high dimensions are not trivially degenerate, not converging to delta functions.

We had already investigated on this potential issue before our first submission. Our analysis is not affected by any apparent issue of degeneracy due to the large dimensionality of the embedding space. To support this statement, in Fig. R1, we display the distribution of the cosine similarity

for all pairs of nodes in two real-world networks: the citation networks Cora and Citeseer. Cosine similarity is measured for embeddings obtained with the node2vec algorithm at different values of the embedding dimension. As the figure clearly shows, the distribution narrows as the embedding dimension increases. However, the width of the distribution does not vanish in the limit of large values of the embedding dimension. Instead, it plateaus to a finite value. As a term of comparison, we consider the distribution of the cosine similarity for random clouds of points in d -dimensional hyper-cubes. In such a case, the distribution tends to be a delta distribution in the limit of large d values. Specifically, the standard deviation of the distribution vanishes as $1/\sqrt{d}$, as predicted by the central limit theorem.

Figure R1: **a** Distribution of the cosine similarity between pairs of nodes of the Cora citation network. We display results valid for node2vec embeddings in $d = 2$, $d = 32$ and $d = 1,024$ dimensions. **b** Same as in panel a, but for the Citeseer citation network. **c** Average value of the cosine similarity distribution of node2vec embeddings as a function of the embedding dimension. We display results for both the Cora and Citeseer citation networks. As terms of comparison, we also display result valid for clouds of points randomly scattered in d -dimensional hyper-cubes. The number of points considered to form the clouds are identical to those of the real-world networks. **d** Same as in panel c, but for the standard deviation of the cosine similarity distributions.

We apologize for not reporting on this result in the previous version of our paper. We included Fig. R1 in the revised version of the SI. The finding is commented in the revised version of the main manuscript, see footnote 2.

2. Eq.6 used in the definition of the optimal embedding dimension does not look correct unless L_∞ is exactly zero. This is because at $d = d_r$, $L(d)$ must be zero, but it is not zero according to eq.6. Also, what is $L(d)$ for $d \gg d_r \gg 1$? In any case, $L(d)$ converging to a not-exactly-zero constant at $d \rightarrow \infty$ is an unexplained mystery.

Thanks for this criticism. The previous version of the paper was not precise enough to clarify the various issues raised by the reviewer in this comment. In the revised version, we carefully expanded the description of how we use Eq.(6), as well as the meaning of the quantities d_r and L_∞ . We summarize below the logic of our argument.

In our paper, we tacitly assume that perfect embedding is surely possible for $d = N$. An example of such a perfect embedding (i.e., lossless representation) is trivially given by the adjacency matrix of the graph, with each row of the matrix containing the coordinates of the corresponding node in the space. Please note that we are not assuming that an (alternative) perfect embedding is not possible for $d < N$. However, we do not a priori know if such a perfect representation is possible for an arbitrary network.

As perfect embedding is surely possible for $d = N$, such a value represents our reference dimension to evaluate the quality of embeddings obtained in lower-dimensional spaces, i.e., $d_r = N$. Unfortunately, for $N \gg 1$ computational limitations do not allow us to use $d_r = N$. As written in the paper, we adopted the following procedure: "... we set the reference value $d_r = N$ and for graphs with more than $N = 500$ nodes, we set $d_r = 500$." We verified that, for our results, the actual "choice of the reference value d_r is not so crucial as long as d_r is large enough (see SI)."

Given the meaning we give to d_r , we evaluate the ability of an embedding method to well preserve network structure only for $d < d_r$. Eq.(6) is indeed used to fit data points corresponding to embedding dimensions in the range $[1, d_r)$. In this range, it is legitimate to expect some loss of information due to the compression of the embedding in a lower-dimensional space. The fact that $L_\infty > 0$ reflects such an information loss.

While working on the revisions of the manuscript, we also realized that we were not properly centering the embedding matrix V . Now, we center the embedding matrix V so that $\sum_{i=1}^N V_{i,s} = 0$, for all $s = 1, \dots, d$. Please note that this operation is important because translation invariance does not hold for the cosine similarity metric. All results of the paper have been updated accordingly. We are happy to report that our main findings are not affected much by the change, meaning that our previous centering operation was already sufficiently accurate.

3. The link prediction part is very unusual in many respects, and needs to be either expanded with extensive justifications or, better, made conventional. The most objectionable part is to call link ij present if $p_{ij} > 0.5$. If this is how a network is constructed, then it is surely very different from the original network! If the path leading to p_{ij} is followed (which has its own many issues), then the standard practice is to rank all ij according to p_{ij} and report the standard metrics - AUPRC, AUC-ROC and similar. AUC is the area under the curve, and the higher the value, the better the classifier.

Thanks for this constructive advice. We changed the link prediction metric from accuracy to the

AUC score (the area under a receiver operating characteristic curve). Updated results are qualitatively similar to the previous ones.

We still learn the vector θ of Eq.(6) to improve the AUC score in the link prediction task. In absence of such a learning step, i.e., setting identical and positive components for the vector θ , then the quantity p_{ij} in Eq.(6) would be equivalent to the cosine similarity between the embeddings of the nodes i and j .

Some other suggestions of less importance.

4. The discussion of normalization of $V^{a,b}$ is missing around eqs.2,3. If they are not normalized, the distances C_{ij} can exceed 1. If they are properly normalized, L cannot exceed 1 (“2” after eq.3 is then a typo).

The definition of cosine similarity is such that $-1 \leq C_{ij} \leq 1$. The maximum value of the rhs of Eq.(3) is thus 2, as stated in the text.

5. It is not mentioned that eq.4, contrary to eq.3, applies only to embeddings of the same dimension.

Thanks, we fixed the issue.

6. A more representative set of citations in machine learning, other than 18,19, could be composed to support the claim that the embedding dimension of many networks is known to be low.

Thanks for pointing out this shortcoming. We have expanded the list of references to make our statement more compelling than it was. Specifically, we added the Recurrent Neural Networks (RNNs) based representation algorithm GraphRNA [3], Generative Adversarial Networks (GANs) based algorithm ARVGA [6] and Graph Convolutional Networks (GCNs) based algorithm GraphSAGE [1] and GCN [4] to support our claim that the embedding dimension of many algorithms are known to be low.

List of changes

To ease the re-evaluation of the manuscript, we have highlighted all changes with red fonts. The major changes we performed are:

- As suggested by both reviewers, we revised the presentation of our results. Major changes were performed in the sections results and conclusion.
- We expanded the set of real networks analyzed. We now consider 135 empirical graphs, some with size $N > 10^6$.
- We included the analysis of two additional graph embedding methods, GraphSAGE and GraphRNA. We validated our approach to the identification of their optimal embedding dimension by performing graph clustering and node classification as downstream tasks.
- We followed the advice of both the reviewers and expanded our list of references.
- We included a new analysis showing that the saturation of the normalized loss function in high-dimensional space is not due to the degeneracy of the cosine similarity metric.

Bibliography

- [1] William L. Hamilton, Rex Ying, and Jure Leskovec. Inductive representation learning on large graphs. In *NIPS*, 2017.
- [2] Xiao Huang, Jundong Li, and Xia Hu. *Accelerated Attributed Network Embedding*, pages 633–641.
- [3] Xiao Huang, Qingquan Song, Yuening Li, and Xia Hu. Graph recurrent networks with attributed random walks. In *SIGKDD Conference on Knowledge Discovery*, 2019.
- [4] Thomas N Kipf and Max Welling. Semi-supervised classification with graph convolutional networks. *arXiv preprint arXiv:1609.02907*, 2016.
- [5] Abhishek Kumar, Piyush Rai, and Hal Daumé. Co-regularized multi-view spectral clustering. *NIPS'11*, page 1413–1421, Red Hook, NY, USA, 2011. Curran Associates Inc.
- [6] Shirui Pan, Ruiqi Hu, Guodong Long, Jing Jiang, Lina Yao, and Chengqi Zhang. Adversarially regularized graph autoencoder for graph embedding. In *IJCAI*, pages 2609–2615, 2018.

Reviewers' Comments:

Reviewer #2:

Remarks to the Author:

I thank the authors for the additional information and revision on the manuscript.

The additional experimentation is appreciated and it addresses the concern on the real-world implications. Under the current framing, I am still not fully convinced by the authors' arguments on the definition of "optimal dimension". The actual problem tackled in the manuscript is in fact low-rank approximation. While on the surface it sounds like a mere rephrasing of the formulation, it emphasises the fact that the manuscript is not considering the underlying noise on the realisation/estimation of the network. Without clearly stating the context, the notion of "optimal dimension" will be misleading.

Reviewer #3:

Remarks to the Author:

The paper has improved in many ways compared to the originally submitted version, and while there are still some concerns remaining, they are relatively minor and can be skipped, so that the paper can be published in its current revision.

Reviewer #2

I thank the authors for the additional information and revision on the manuscript.

The additional experimentation is appreciated and it addresses the concern on the real-world implications. Under the current framing, I am still not fully convinced by the authors' arguments on the definition of "optimal dimension". The actual problem tackled in the manuscript is in fact low-rank approximation. While on the surface it sounds like a mere rephrasing of the formulation, it emphasises the fact that the manuscript is not considering the underlying noise on the realisation/estimation of the network. Without clearly stating the context, the notion of "optimal dimension" will be misleading.

We thank the referee for the appreciation of our efforts. We are grateful for all the constructive criticisms that the reviewer provided. They truly helped us to substantially improve the quality of our work.

Regarding the remaining concern, we thank the reviewer for making this nice connection. We agree on the fact that we are dealing with a low-rank approximation. As a matter of fact, we seek the low-dimensional (i.e., low-rank) approximation that best describes the high-dimensional (i.e., full-rank) embedding matrix. We differentiate from the standard setting considered in low-rank approximation problems in two main respects.

First, we are not exactly relying on a standard metric of distance (e.g., Frobenius distance) between the low- and the full-rank embedding matrices. Instead, we have a measure of collective congruence between the embeddings. There is a perfect match between the two metrics as long as we are dealing with spectral embedding methods, e.g., Laplacian Eigenmaps. In that case in fact, we are relying on the decomposition of the full-rank operator in principal eigencomponents which corresponds to the minimization of the distance between the low- and full-rank matrices. However, we are not in the position to tell if such an equivalence holds for other embedding methods too.

Second and more important, there is a distinction, both in terms of variables and constraints, between the optimization problems tackled in standard low-rank approximations and ours. In a standard low-rank approximation, one generally aims at finding the best low-rank reduction of a full-rank matrix by constraining the rank of the approximation. In our case, we focus on the problem of finding the best rank reduction constrained by a maximum amount of tolerated discrepancy between the low- and the full-rank representations.

Given the above considerations, we prefer to avoid to use the expression low-rank approximation in framing our work. Nevertheless, we added a few sentences at page 8 of the revised version of the manuscript to stress the main difference between the optimization problem that we consider and the one studied in standard low-rank approximations. We like very much the connection made by the reviewer, and we believe it represents an additional improvement for our work.

Also at page 8 of the revised version of the manuscript, we included a sentence to state that the considered approach does not account for the fact that the network at hand may be a specific realization of some underlying stochastic model which the embedding method may rely on.

Finally, we understand the concern raised by reviewer regarding the use of the expression "optimal

dimension.” We originally decided to use it in accordance with the papers by Yin *et al.*, i.e., Refs. 20 and 35, which represent the foundations of our work. However, we agree with the reviewer on the fact that one may associate the expression optimal dimension to the existence of a maximum value of some metric of quality or performance. Instead, we observe a saturation. To fully address the reviewer’s concern we removed the adjective “optimal” from the paper, and replaced it with mild adjectives such as “proper,” “appropriate” and “suitable.” After the definition of Eq. (5), we mainly refer to the quantity simply using the notation $d_o(\epsilon)$.

We hope that the reviewer will appreciate our changes/additions, and provide a positive recommendation for the publication of the manuscript.

Reviewer #3

The paper has improved in many ways compared to the originally submitted version, and while there are still some concerns remaining, they are relatively minor and can be skipped, so that the paper can be published in its current revision.

We thank the referee for the very positive comment. We believe that our work significantly benefitted from the pertinent and constructive criticisms raised in the first round of review.

List of changes

To ease the re-evaluation of the manuscript, we have highlighted all changes with red fonts. The major changes we performed are:

- We included a new paragraph at page 8 of the main text. Here, we describe the relation between standard low-rank approximations and our problem. We further give explicit context to the formulation of our problem, as requested by reviewer #2.
- We removed the expression “optimal dimension” in every part of the paper. Title and abstract of the paper have been changed accordingly.

Reviewers' Comments:

Reviewer #2:

Remarks to the Author:

Thanks for the response and changes to the manuscript.

My primary concern, which involves the definition of optimality, has largely been addressed. The possibility of potential confusion has been reduced by the change of the wordings. The current version of the manuscript generally looks good to me.